# High Genetic Diversity in Third-Generation Cephalosporin-Resistant *Escherichia coli* in Wastewater Systems of Schleswig-Holstein

**DOI:** 10.3390/pathogens13010090

**Published:** 2024-01-20

**Authors:** Laura Carlsen, Matthias Grottker, Malika Heim, Birte Knobling, Sebastian Schlauß, Kai Wellbrock, Johannes K. Knobloch

**Affiliations:** 1Institute of Medical Microbiology, Virology, and Hygiene, Department for Infection Prevention and Control, University Medical Center Hamburg–Eppendorf, Martinistraße 52, 20246 Hamburg, Germany; la.carlsen@uke.de (L.C.); b.knobling@uke.de (B.K.); 2Laboratory for Urban Water and Waste Management, Technische Hochschule Lübeck, University of Applied Sciences, Mönkhofer Weg 239, 23562 Lübeck, Germany; matthias.grottker@th-luebeck.de (M.G.); sebastian.schlauss@ebhl.de (S.S.); kai.wellbrock@th-luebeck.de (K.W.)

**Keywords:** wastewater treatment plants, multi-resistant bacteria, antibiotic resistance, β-lactamase, *E. coli*

## Abstract

The spread of multidrug-resistant bacteria from humans or livestock is a critical issue. However, the epidemiology of resistant pathogens across wastewater pathways is poorly understood. Therefore, we performed a detailed comparison of third-generation cephalosporin-resistant *Escherichia coli* (3GCREC) from wastewater treatment plants (WWTPs) to analyze dissemination pathways. A total of 172 3GCREC isolated from four WWTPs were characterized via whole genome sequencing. Clonal relatedness was determined using multi-locus sequence typing (MLST) and core genome MLST. Resistance genotypes and plasmid replicons were determined. A total of 68 MLST sequence types were observed with 28 closely related clusters. Resistance genes to eight antibiotic classes were detected. In fluoroquinolone-resistant isolates, resistance was associated with three-or-more point mutations in target genes. Typing revealed high genetic diversity with only a few clonal lineages present in all WWTPs. The distribution paths of individual lines could only be traced in exceptional cases with a lack of enrichment of certain lineages. Varying resistance genes and plasmids, as well as fluoroquinolone resistance-associated point mutations in individual isolates, further corroborated the high diversity of 3GCREC in WWTPs. In total, we observed high diversity of 3GCREC inside the tested WWTPs with proof of resistant strains being released into the environment even after treatment processes.

## 1. Introduction

The continuous release of antibiotic-resistant bacteria by wastewater treatment plant (WWTP) effluents after the treatment process to the urban water pathways is considered a critical One Health issue. The One Health concept focuses on the link between humans, animals, and the environment on a local and global scale to achieve better community health [1]. This concept includes the emergence, development, and spread of antibiotic-resistant microorganisms. The microbial link between humans, animals and the environment poses a global health risk as resistance can accumulate and adapt to different niches [1]. WWTP effluents are considered one of the most relevant sources for the introduction of resistance genes and multi-resistant bacteria into adjacent waters and ecosystems [2]. Increasingly, multi-resistant bacteria are introduced into the water cycle through the input of human, and animal feces and may be further selected within bacterial populations in the aquatic ecosystem through the introduction of antibiotics from different origins. Several studies have shown that interactions between antibiotics and the development of resistance in aquatic ecosystems appear [3,4]. 

Previous studies demonstrate that third-generation cephalosporin-resistant *Escherichia coli* (3GCREC) and other resistant *Enterobacterales* can be detected in municipal wastewater and even surface water worldwide [2]. In Germany, both resistant *E. coli* isolates carrying various resistance genes and even carbapenem-resistant *Enterobacterales* (CRE) have already been detected in the effluent of wastewater treatment plants [5,6,7]. In some cases, these resistance genes are still detected after post-treatment by a fourth treatment step (ozone treatment) and were also found in adjacent water systems [2,8]. Even an increase in the abundance of multi-resistant bacteria, including 3GCREC, in wastewater treatment plants was demonstrated in some studies, posing a potential risk not only to the immediate environment and adjacent water systems but also to WWTP workers [9,10]. 

In addition to resistance to third-generation cephalosporins, further resistance to antibiotic groups like fluoroquinolones is also becoming increasingly important [11]. Fluoroquinolones such as ciprofloxacin are usually given for infections with bacteria with existing resistance to third-generation cephalosporins, so additional resistance in this direction poses a further health risk [12]. In this regard, in some cases, specific resistance genes are not even necessary for bacteria to develop phenotypic resistance to fluoroquinolones. Several point mutations in the genes encoding the target proteins for fluoroquinolone antimicrobial activity are now known to cause resistance in *E. coli* and other species, the majority of which affect the *gyrA, parC,* and *parE* genes [13,14]. Genes mediating resistance to fluoroquinolones, as well as resistance-mediating point mutations, have already been detected in the influent and effluent of wastewater treatment plants [15].

WWTPs display the last barrier for harmful substances and microorganisms carried via wastewater before being released into nearby water systems like rivers and lakes. The release of resistant bacteria or resistance genes into the environment can be traced back to the human and animal food chain, reintroducing critical resistance into human and veterinary medicine [3,6]. For this reason, the WWTP influent and effluent are increasingly being analyzed for resistance genes and multidrug-resistant bacteria using various methods in order to understand the spread of antimicrobial resistance between humans, animals, and the environment in the spirit of the One Health concept. However, it is not yet understood how bacterial clonal lineages behave during the wastewater treatment process and whether specific clones can individually accumulate or persist within WWTPs. Therefore, in this study, 3GCREC from four representative WWTPs in northern Germany were characterized on a large scale from both influent and effluent-treated wastewaters via whole genome sequencing and were investigated for clonal relatedness and the presence of resistance determinants as well as plasmid replicons.

## 2. Materials and Methods

### 2.1. Selection of Representative Wastewater Treatment Plants

Four municipal wastewater treatment plants were selected whose treatment processes represented a large part of the existing wastewater treatment plants in Schleswig-Holstein and which were located in different regions of the federal state. The selection was made in such a way that after the biological stage (Acc. to the activated sludge process), further processing stages (e.g., trickling filters, fixed bed reactors, polishing lagoons) were operated before effluent wastewater was released into receiving water bodies. Detailed information on each WWTP is provided in Appendix A. 

Sampling, identification, and antimicrobial susceptibility testing An investigation of the individual WWTP was performed at different time periods during the study period from 20 July 2017, to 27 September 2017. During seven-day periods for each selected WWTP, 24 h composite samples of influent water (raw sewage) as well as effluent water were taken using Buehler 3010 stationary automatic water samplers (Hach, Düsseldorf, Germany) at days 1, 2, 5, and 6, as well as days 2, 3, 6 and 7, respectively. Each 24 h composite sample consisted of 96 individual 75 mL samples taken every 15 min. The shift of 24 h between influent and effluent wastewater samples was used to mimic the mean hydraulic retention time of wastewater within the treatment process. 

Preliminary experiments were carried out to test which quantities of the different types of water should be used on selective media to obtain single colonies for the generation of pure cultures. For influent wastewater, 100 μL was plated directly on chromID^®^ ESBL agar (bioMérieux, Marcy l’Etoile, France), whereas for effluent wastewater, 50 mL was enriched via filtration using a 0.45 μm cellulose-mixed ester filter with a 47 mm diameter (EZ-PAK 020, Merck Milipore, Darmstadt, Germany) which was subsequently placed on a chromID^®^ ESBL agar (bioMérieux, Marcy l’Etoile, France). Inoculated media were incubated at 42 °C for 24 h to select the growth of *E. coli*. Pure cultures of up to ten randomly selected β-glucuronidase-producing, red-growing colonies from each sample were generated and identified using MALDI-TOF mass spectroscopy and a Microflex LT mass spectrometer (Bruker Daltonics, Bremen, Germany) and the BioTyper^TM^ database (Bruker Daltonics, Bremen, Germany). From plates with less than 10 β-glucuronidase-producing colonies, all available colonies were isolated. Antibiotic susceptibility testing was performed using agar diffusion assays with discs containing ciprofloxacin (CIP, 5 μg), gentamycin (CN, 10 μg), tetracycline (TE, 30 μg) and trimethoprim (TMP, 5 μg) and using a double-disc diffusion assay with discs containing ceftriaxone (CRO, 30 µg), amoxicillin/clavulanic acid (AMC, 30 µg) and ceftazidime (CAZ, 30 µg, all discs by Becton Dickinson, Heidelberg, Germany). Inhibition zone diameters were evaluated according to the European Committee on Antimicrobial Susceptibility Testing (EUCAST clinical breakpoints, v 7.1) limits. Isolates chosen for sequence analysis with a negative double disc diffusion assay result were further analyzed using a VITEK^®^ 2 antibiotic susceptibility testing (AST) system (version 9, bioMérieux, Marcy l’Etoile, France) with the antibiotic card type AST-428 [16,17,18]. Isolates with VITEK^®^-confirmed resistance to third-generation cephalosporins were included in the study. All 3GCREC isolates were assigned to one of the 18 observed resistance phenotypes (Appendix A). 

### 2.2. Whole Genome Sequencing and Data Analysis

The selection strategy for representative isolates for detailed molecular analysis was to choose one isolate from each resistance type per day and sampling site, if available. Bacterial genomic DNA from pure cultures was purified using the QIAsymphony^®^ DSP Virus/Pathogen Mini Kit (Qiagen, Venlo, The Netherlands) and a QIAsymphony^®^ (Qiagen) instrument. Chromosomal DNA was sheared using a Bioruptor^®^ Pico device (diagenode, Seraing, Belgium). DNA library preparation for sequencing was performed using the NEBNext Ultra DNA Library Prep Kit for Illumina (NEB, Ipswich, MA, USA) and NEB Next Multiplex Oligos for Illumina. The concentration of the library pool was measured again using Qubit, and the fragment length distributions of the final libraries were analyzed with the DNA High Sensitivity Chip on an Agilent 2100 Bioanalyzer (Agilent Technologies, Santa Clara, CA, USA). The library pool was normalized to 2 nM and sequenced on the NextSeq500 (Illumina, San Diego, CA, USA) with 2 × 151 bp in the mid-output format. Samples had 1.1 to 4.6 mio reads per sample. Detailed information about sequencing quality can be found in Appendix A. 

The assembly of sequenced genomes was performed using a Velvet assembler integrated in SeqSphere software (version 1.1.04) (Ridom, Münster, Germany). Analysis via classical multi-locus sequence typing (MLST) using the Warwick scheme [19] and core genome multi-locus sequence typing (cgMLST) with the published cgMLST [20,21,22] scheme was automated after assembly in the software. The threshold for identical isolates was set at 10 different alleles according to the guidelines of Ridom SeqSphere+ [23]. To ensure international comparability, the sequence types of the classical Warwick MLST scheme were used for strain designation. The sequences of all isolates are available at BioProject Nr. PRJNA832938.

The sequences were uploaded to the ResFinder [24,25,26] pipeline of the Center for Genomic Epidemiology (CGE) [27] to identify the presence of resistance genes and point mutations for fluoroquinolones in all isolates, as fluoroquinolone resistance is known to be mediated by point mutations as well [13,14]. For isolates lacking a detectable resistance determinant for third-generation cephalosporins, VITEK^®^ 2 AST was additionally performed to confirm 3GCREC. All isolates that formed genetically and closely related clusters were tested for their plasmid replicons using the CGE pipeline PlasmidFinder [26,28]. Descriptive statistics were performed using R (version 3.6.2) and RStudio (version 1.2.5033; posit PBC, Boston, USA) with the library *dplyr* activated [29]. For statistical analysis, isolates were grouped according to the presence of one or more resistance determinants (groups were as follows: none, single mutation, double mutation, multiple mutations [≥3], resistance gene, as well as resistance gene with single, double or multiple mutations (Appendix A). To assess the association between the presence of these determinants and the fluoroquinolone-resistant phenotype, Fisher’s exact test was performed using R (version 4.3.1 [30]) and R Studio (version 2023.12.0369; posit PBC, Boston, USA [31]) with the rstatix package activated [32]. First, the command *fisher_test* was applied, followed by the post hoc *pairwise_fisher_test* in cases where the *p*-value of the *fisher_test* was significant (*p* < 0.05). The Benjamini–Hochberg method was used for correction [33].

## 3. Results

A total of 307 representative β-glucuronidase-positive isolates (WWTP 1: n = 78, WWTP 2: n = 80, WWTP 3: n = 67, WWTP 4: n = 80) were randomly isolated from chromID^®^ ESBL agar. In WWTPs 1 and 3, only 39 and 28 β-glucuronidase-producing colonies were observed in effluent wastewater samples, respectively. Each one isolated from the influent samples of WWTP 1 and WWTP 3 was identified as *Citrobacter freundii* despite the presence of a β-glucuronidase and was, therefore, excluded from further evaluation. Phenotypic screening of the enclosed 305 *E. coli* revealed additional resistance against trimethoprim, tetracycline, fluoroquinolones (ciprofloxacin), and aminoglycosides (gentamycin) with 154 (50.5%), 153 (50.2%), 83 (27.2%) and 80 (26.2%) isolates displaying resistant phenotypes, respectively. A positive double disc diffusion assay indicating ESBL expression could be observed in 294 (96.4%) isolates. In 11 isolates, no enlargement of the inhibition zone of the two tested third-generation cephalosporins was observed. Depending on the combination of phenotypic resistance, 3GCREC could be divided into 18 different resistance types (R01 to R18, Appendix A). Several resistance types were observed in all tested water samples (Figure 1). The resistance type R15 (the double disc diffusion assay was positive but lacking in additional resistance in 3GCREC) was the most common (n = 70) and also the most frequent phenotype in WWTPs 1 (n = 15), 2 (n = 20) and 4 (n = 16) and the second most common phenotype in WWTP 3 (n = 19). Most frequently, a combined resistance against two antibiotic groups in addition to third-generation cephalosporin resistance was detected against trimethoprim and tetracycline (n = 36). Instances of 3GCREC isolate with further resistance to all antibiotic groups tested in the agar diffusion assay (R01) were found 32 times. 

In total, 172 3GCREC isolates were included in the genomic analysis (WWTP 1: n = 49; WWTP 2: n = 39; WWTP 3: n = 33; WWTP 4: n = 51). In the sequenced representative isolates, known β-lactamase resistance genes could be detected in 158 (91.9%) isolates (Figure 2). Among the ß-lactamase genes, the *bla*_CTX m_ ESBL family was observed most frequently (n = 146) with *bla*_CTX-M-15_ (n = 69), *bla*_CTX-M-1_ (n = 43), *bla*_CTX-M-14_ (n = 19), *bla*_CTX-M-27_ (n = 6), *bla*_CTX-M-9_ (n = 4), *bla*_CTX-M-14b_ (n = 4) and *bla*_CTX-M-55_ (n = 1). AmpC encoding genes (*bla*_CMY_) could only be observed in two isolates. No carbapenemase genes were observed. Resistance against third-generation cephalosporins was confirmed using VITEK^®^ 2 AST (bioMérieux, Marcy l’Etoile, France) in isolates lacking detectable known resistance mechanisms. Seven of all sequenced 3GCREC isolates did not have any detected resistance genes at all. 

Aminoglycoside resistance genes were observed in 124 (72.1%) isolates, with sulfonamide resistance genes in 113 (65.7%) isolates. Resistance genes for resistance to trimethoprim or tetracycline were found in 89 (51.7%) or 78 (45.3%) isolates, respectively. Genes encoding a resistance mechanism against MLS (macrolides, lincosamides, and streptogramins), fluoroquinolones, and phenicols were evident in 51 (29.7%), 47 (27.3%) and 33 (19.2%) of the isolates, respectively. In total, 86.5 to 100% of the isolates carrying aminoglycoside, tetracycline, or trimethoprim resistance genes also showed phenotypic resistance to the tested antibiotics. In contrast, of the isolates with fluoroquinolone resistance-associated genes, 76.6% showed no phenotypic resistance to ciprofloxacin. Of the 51 isolates displaying multiple mutations (8 were combined with an additional resistance-associated gene), a fluoroquinolone-resistant phenotype was observed in 50 (98.0%) (Appendix A). By contrast, in 74 isolates in the groups of single or double mutations with or without the presence of an additional resistance-associated gene, only four isolates (5.4%) displayed phenotypic resistance. In isolates lacking resistance determinants (n = 47), no resistant phenotype was observed. The association of a resistant phenotype with multiple mutations was statistically significant (Appendix A). Among the observed mutations, the most frequently occurring combination of point mutations (n = 35) was *parC*(S80I), *gyrA*(S83L), and *gyrA*(D87N), with an additional varying point mutation of *parE* in 23 of the isolates. 

The typing of sequenced isolates using cgMLST displayed the wide diversity of the isolates (Figure 3) with several subclusters, even within the most frequent classical MLST sequence types (STs). In total, 68 different STs (Appendix A) were detected, with 35 STs observed only once (singleton sequence types; SSTs). ST15488 was newly characterized with one isolate in this study, representing a single locus variant of ST100, which is already known as an antibiotic-resistant ST in swine farms in Europe [34,35]. The most common STs identified were ST10 (n = 16), ST38 (n = 14), ST131 (n = 12) and ST48 (n = 10). Several sequence types were found to be single locus variants (SLVs) to ST10 and ST38 and were further described as the ST10 and ST38 complex, respectively. Only a few closely related isolates, as determined using cgMLST (less than 10 different alleles; Appendix A), were identified in different WWTPs. Five subclusters were identified to have clonal-related isolates in two different WWTPs within the ST10 complex (two subclusters), as well as in ST48, ST131, and ST58, although the isolates in ST58 were completely identical. Two subclusters within the sequence types ST38 and ST69 were identified in three different WWTPs. There were seven other pairs of genetically and closely related isolates, but each came from the same WWTP. In contrast, other isolates belonging to the same classical ST sometimes showed very strong genetic differences using cgMLST with up to 1051 different alleles in ST10 (Appendix A). In most cases, the closely related isolates from the individual wastewater treatment plants originated from the influent and associated effluent (Appendix A). Isolates from the ST10 complex were found in all four WWTPs, with four isolates (ST209) of two different WWTPs being closely related. Isolates from different WWTPs were also determined to be clonally and closely related within the ST38 complex and in ST131. Isolates of both sequence types were also found in all four WWTPs. The three sequence types, ST48, ST783, and ST9962, all showed very low genetic diversity within their STs, according to cgMLST. ST48 occurred only in WWTP 2, displaying two different resistance gene clusters. ST783 and ST9962 were found exclusively in WWTP 1 and WWTP 3, respectively. For ST9962, exactly one sample was found on three of the four sampling days in the influent and the associated effluent on the following day (Appendix A). Also, within the other sequence types, no remarkable separation in the influent and effluent samples was recognizable. 

Resistance genes within ST10 are highly diverse (Appendix A). Of the ten phenotypically fluoroquinolone-resistant isolates, only three had a resistance gene to fluoroquinolones, but all had the combination of point mutations parC(S80I), gyrA(S83L), and gyrA(D87N). ST38 also shows diverse resistance genes, while ST131 can be divided into two groups of resistance gene patterns. Both STs have combinations of point mutations in the phenotypically fluoroquinolone-resistant isolates without fluoroquinolone-resistance genes. In ST48, two different combinations of resistance genes showed up, with five of the isolates having resistance genes to fluoroquinolones but also showing up as phenotypically non-resistant. All isolates with resistance genes to fluoroquinolones also possessed resistance genes to sulfonamides and trimethoprim. In both ST783 and ST9962, resistance genes are very homogeneously distributed, with ST9962 having fluoroquinolone resistance genes throughout but phenotypically showing up as non-resistant. 

Within the ST10 complex, two clusters were formed from ST10 isolates, as well as two further clusters from the SLVs ST209 and ST44. The isolates of both clusters of ST10 showed homogeneous plasmid replicons (Appendix A). While four different replicons were detected in the cluster from WWTP 2, the isolates from the WWTP 4 cluster had only one replicon each. The ST209 cluster also showed a difference between the isolates from WWTP 4 with three replicons each and the isolates from WWTP1 with one additional replicon. Of the two isolates from the ST44 cluster, one possessed one more replicon than the other. Within the ST38 complex, there were two clusters. For the ST38 isolates, with one exception, no replicons could be identified, and the two ST2003 isolates differed in one replicon. In ST131, two different clusters appeared, including one cluster with two isolates from different WWTP homogeneous replicons, while within the WWTP 1 cluster isolates, there was a difference in one replicon. The WWTP 2-specific ST48 formed an entire cluster of all isolates, which could be divided into two clearly distinct groups based on the replicons. In four isolates, which were all genetically identical, the same six replicons were identified, while in all remaining isolates, only two replicons could be detected. In the two sequence types ST58 and ST69, three clusters each could be identified; with one exception, these were homogeneous but different from the other clusters. The clusters of ST783 and ST9962, with one exception, each possessed no or homogeneous replicons, respectively. The isolates of the clusters ST46, ST93, ST182, ST226, ST541, and ST3873 each had different replicons, but the clusters consisted of only two or three isolates. All other small clusters with two isolates each belonged to ST90, ST120, ST216, ST410, ST453, ST2067, and ST4981, and each had homogeneous replicons. 

## 4. Discussion

In this study, the clonal relation and content of genetic resistance determinants in third-generation cephalosporin-resistant *E. coli* (3GCREC) were analyzed in comparison to isolates from the influent wastewater and purified effluent wastewater. During the isolation of β-glucuronidase-producing, red-growing colonies, the species *C. freundii* was observed in less than 1% of isolates, as can be expected in rare events [36,37]. By contrast, recently, a high proportion of β-glucuronidase-producing third-generation cephalosporin-resistant *C. freundii* was described in hospital-specific wastewater [38,39]. 

The most common phenotypic resistance, in addition to phenotypic β-lactam resistance in 3GCREC, was observed against trimethoprim and tetracycline, most frequently in combination with each other. Both groups of antibiotics are used in human and veterinary medicine [40,41]. All four WWTPs were located in rural regions with livestock farming, but hospitals were also located in the catchment area of three WWTPs, so the origin of these resistances could not be clearly assigned.

Of the 172 isolates sequenced, a β-lactamase gene was identified in only 158, although all isolates were phenotypically 3GCREC. This can be justified by the fact that *E. coli* can also display a resistant phenotype against β-lactams via the differential expression of outer membrane porins [42]. Also, the loss of plasmids between phenotypic resistance testing and the sequencing of the isolates cannot be fully excluded. Other studies demonstrated an increased load of multiple resistant 3GCREC and even carbapenem-resistant *Enterobacterales* (CRE) in WWTPs [6,43,44]. However, carbapenemase genes could not be detected in any isolate used in our study. Nevertheless, resistance genes against aminoglycosides, sulfonamides, trimethoprim, tetracyclines, MLS, phenicols, and fluoroquinolones were observed, indicating a high diversity in resistance genes in 3GCREC. 

The occurrence of fluoroquinolone resistance is particularly significant as fluoroquinolones such as ciprofloxacin are usually the antibiotic of choice for infections caused by bacteria with existing resistance to third-generation cephalosporins [12]. For this reason, the separate classification of fluoroquinolone-resistant 3GCREC, so-called 3MRGN (triple multidrug-resistant Gram-negative), exists in Germany, and 3MRGN is being monitored [45]. Of the 55 phenotypically fluoroquinolone-resistant isolates, only 12 showed a corresponding resistance-associated gene, while the others (with one exception) showed a combination of point mutations associated with fluoroquinolone resistance. Thereby, statistical analysis displayed a significant association of more than two-point mutations with phenotypic resistance, whereas fluoroquinolone resistance-associated genes alone were not significantly associated with phenotypic resistance as was described previously [46,47,48]. The most commonly found combination of *parC*(S80I), *gyrA*(S83L), and *gyrA*(D87N) is already known to be able to develop phenotypic resistance to ciprofloxacin and has been detected in *E. coli* both upstream and downstream of sewage treatment plants [13,14]. The fact that 39 phenotypic fluoroquinolone susceptible isolates displayed a non-wildtype with already one or two first-step mutations present indicates that a relevant proportion of 3GCREC isolates might develop more rapidly phenotypic resistance under selective conditions using fluoroquinolones [46,47,48]. 

The frequent occurrence of combined resistance genes against different antibiotic groups already indicates the presence of several different plasmids and is confirmed by the finding of various different and multiple replicons, even in closely related isolates. Combinations of β-lactamases and aminoglycoside resistance genes, which have already been described as frequent combinations of antibiotic group resistance [49], were particularly common.

All isolates were analyzed for clonal relatedness using cgMLST as well as classical MLST. The occurrence of a total of 68 different sequence types in 172 isolates indicates high genetic diversity in wastewater-associated 3GCREC. As all four WWTPs have several different entry sources for wastewater, a high diversity of bacteria like *E. coli* in the influent wastewater was expected, yet the diversity of remaining 3GCREC in the effluent samples after the wastewater treatment process was comparatively high. Three sequence types and their respective single locus variants (SLVs) were identified in all of the investigated WWTPSs (ST131, ST10 complex, ST38 complex). All of these lineages are considered emerging clonal lineages [50] and are probably released into the wastewater from several entry sources, resulting in a wide distribution inside the wastewater systems. 

ST131 is considered the most common extraintestinal pathogenic *E. coli* [51], so its wide distribution in rural Schleswig-Holstein is not surprising. Interestingly, this clone was only identified once in a recent study investigating 90 3GCREC in hospital wastewater in the neighboring non-rural state of Hamburg despite the frequent identification of ST131 in patient isolates [38,39], indicating that ST131 in communal wastewater is more likely introduced from the general population than hospital-specific. In contrast, in Hamburg hospital wastewater, a wastewater-adapted clonal lineage (ST635) was observed as dominant [38], which was not observed in the Schleswig-Holstein WWTP samples. However, it must also be kept in mind that a WWTP has a much larger volume of wastewater than a single hospital and that only single isolates were sequenced. Thus, it cannot be ruled out that ST635 could also be found in a more intensive sequencing of *E. coli* in the WWTPs. In these clonal lineages, the strains could be clearly subdivided by cgMLST, with some isolates displaying a distance of several hundred different alleles (Appendix A), corroborating their high diversity in wastewater. An analysis via classical MLST without consideration of cgMLST, therefore, seems to be insufficient for a detailed analysis of wastewater samples along the water pathways. 

ST10 has been described to be highly diverse [52,53,54]. Analysis with cgMLST revealed two clusters inside ST10 from two different WWTPs with a difference of 69 alleles to each other (Appendix A). Each of these clusters consisted of isolates with homogenous resistance genes and detected plasmid replicons, with the cluster containing more resistance genes also showing more replicons and, thus, giving proof for the uptake or loss of plasmid-mediated resistance genes. However, it is unclear whether these events occur within the wastewater pathways or whether differently-equipped isolates already enter the wastewater. A similar separation could be observed for the ST10-variant ST209 cluster with varying resistance genes and replicons depending on the WWTP. In ST10, the combination of point mutations *parC*(S80I), *gyrA*(S83L), and *gyrA*(D87N) was detected in all ten phenotypically fluoroquinolone resistance-positive isolates as well as in all but two non-ST10 isolates of the ST10 complex, which may indicate that this ST is particularly capable of achieving fluoroquinolone resistance through point mutations. 

Isolates of ST38 were detected in all four WWTPs and showed diverse resistance genes, which fits the description of this sequence type as a global high-risk clone [55]. Interestingly, resistance genes against fluoroquinolones could not be detected in any of the phenotypic fluoroquinolone-resistant isolates, whereas the two isolates containing resistance genes were phenotypically susceptible, meaning that, in this clone, the phenotypical resistance to fluoroquinolones seems to be more likely associated with point mutations than with resistance genes. Despite multiple resistance genes inside all ST38 isolates, only for one isolate of the tested clonal clusters could a replicon be identified, meaning the resistance genes are either located on the chromosome or the existing plasmid replicons could not be detected. 

Sequence types ST48, ST783, and ST9962 were identified for several days only in WWTP 2, WWTP 1, and WWTP 3, respectively. ST48 has been described as pathogenic to humans and animals and has been found in agriculture [56,57], so it can be speculated that there is a specific source of input for this ST in the catchment of the WWTP. All isolates originated from the same WWTP and are genetically closely related but can be divided into two clearly delineated resistance gene/replicon groups, so plasmid uptake or loss can be assumed. As both detected resistance gene patterns and replicon combinations were found in the influent and effluent of the WWTP, the causes of genetic events seem to take place upstream of the wastewater treatment process. ST783 has already been described in cattle and sheep [58]. Even if usually no agricultural wastewater should enter municipal wastewater pipes, an exceptional entry cannot be completely ruled out. It has already been demonstrated that there are clonal lineages of *E. coli* that can persist in wastewater permanently without any apparent permanent source of entry [38]. Accordingly, it is possible that even the single entry of a clone into a wastewater pipe upstream of the wastewater treatment plant can lead to this clone remaining there permanently and being released into the wastewater over long time periods. ST9962 is an SLV of *E. coli* ST738, which was also described in animals [59]. For ST9962 and ST48, it is notable that both STs consistently did not express a fluoroquinolone-resistant phenotype despite most isolated carried genes associated with fluoroquinolone resistance. This suggests that these STs may not express some resistance. Nevertheless, it is possible that they can act as a reservoir for resistance plasmids. Since all ST48, ST783, and ST9962 were found both in the influent and effluent of the respective WWTPs, neither elimination nor strong enrichment of these STs occurred within the wastewater treatment process. 

Tracing back individual clonal lines to their entry into the wastewater pathways seems to be difficult. However, the sequence types associated with a continuous upstream source of release (ST48, ST783, and ST9962) seem to be more likely associated with livestock, as could be expected for a rural federal state. A high diversity in the presence of resistance determinants has been shown throughout. Particularly striking is the frequent occurrence of point mutations associated with fluoroquinolone resistance, which appear to have resulted in fluoroquinolone resistance in some isolates despite the absence of resistance genes.

## 5. Conclusions

In conclusion, this study revealed a high diversity of 3GCREC in the wastewater treatment plants investigated. In addition to strains being internationally known as emerging clonal lineages, various clonal lineages whose source of entry remains unknown were also detected. Numerous multiple resistance genes against eight different antibiotic groups, as well as various and multiple plasmid replicons, were observed, which also varied in closely related isolates and further corroborated with the diversity of 3GCREC in wastewater treatment plants. No evidence of the accumulation of specific clones within the wastewater treatment process was observed, but resistant organisms were not completely removed, and a diverse pool of resistance genes was released into the environment.

## Figures and Tables

**Figure 1 pathogens-13-00090-f001:**
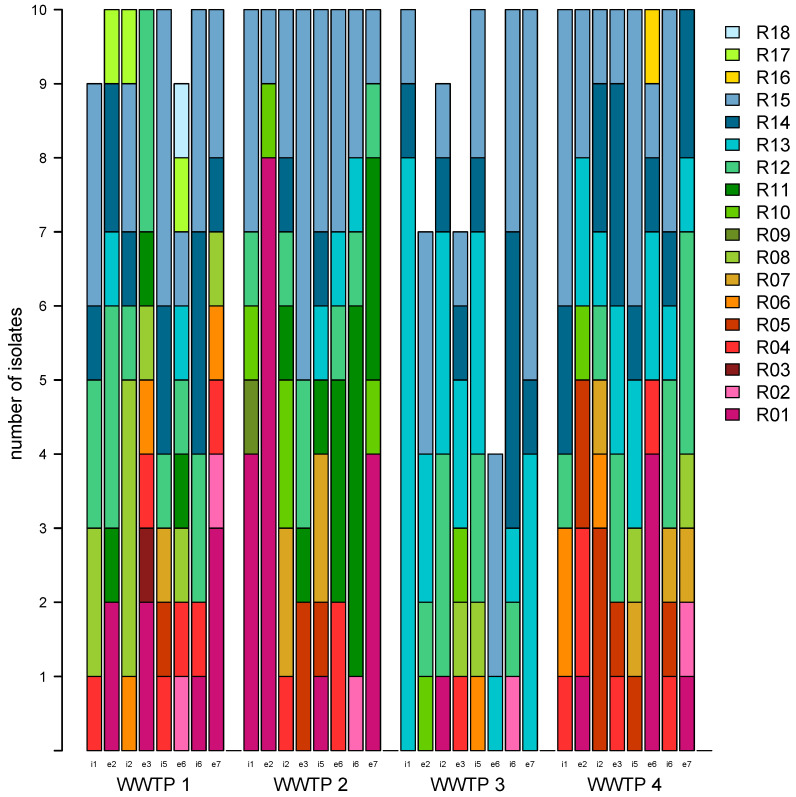
Subdivision of all isolates into phenotypic resistance types. If available, ten isolates per day from each WWTP and sampling site were analyzed for their resistance type (R01–R18). Resistance types displayed in red shades (R01 through R07) show phenotypic resistance to fluoroquinolones. Missing isolates were either identified as non-*E. coli* (influent), or there were no 10 β-glucuronidase-producing colonies (effluent). Letters and numbers mark the sampling location (i: influent; e: effluent) and sampling day within the sampling week.

**Figure 2 pathogens-13-00090-f002:**
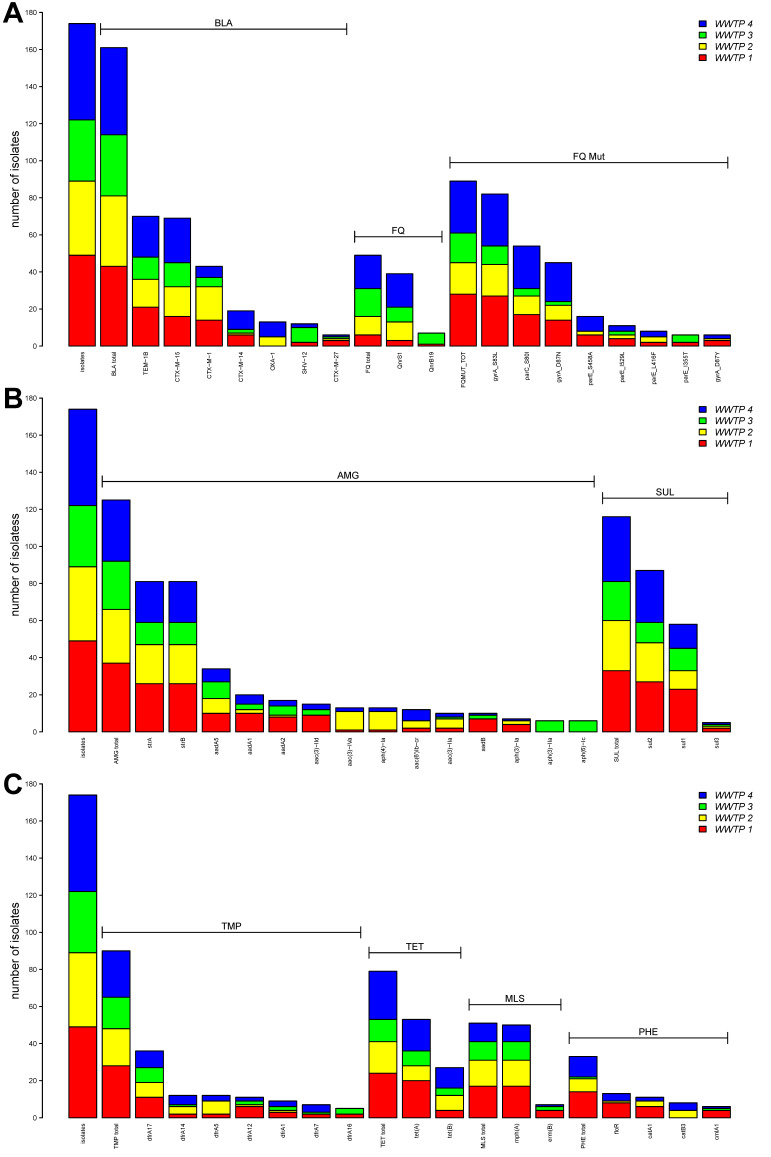
Resistance genes detected in all *E. coli*. Genes encoding resistance against β-lactams, fluoroquinolones (FQ), point mutations associated with FQ resistance (**A**), aminogylcosides (AMG), sulfonamides (SUL) (**B**), trimethoprim (TMP), tetracyclines (TET), MLS (macrolides, lincosamides, streptogramines) and phenicols (PHEN) (**C**) are displayed. Different colors represent the different WWTPs. The first bar represents the total number of analyzed isolates. Only genes/mutations that occurred at least five times are displayed.

**Figure 3 pathogens-13-00090-f003:**
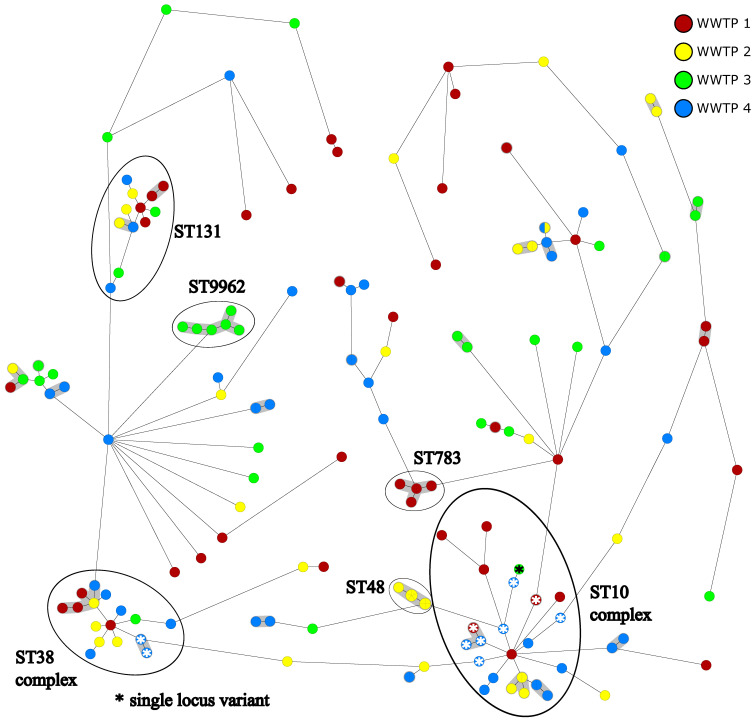
cgMLST analysis of clonal relations in *E. coli* isolates from all WWTPs. Isolates are colored depending on their WWTP origin. Grey-shaded clusters show clonally related subgroups according to cgMLST. Single locus variants of ST10 and ST38 are marked with a star. Additional information about phenotypic resistance types and genetic distances is displayed in Appendix A.

## Data Availability

The sequences of all isolates are available at BioProject Nr. PRJNA832938 at NCBI BioProject.

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
