# Peer review of "High Genetic Diversity in Third-Generation Cephalosporin-Resistant Escherichia coli in Wastewater Systems of Schleswig-Holstein"

_pathogens, 2024, doi:10.3390/pathogens13010090_

Round 1

Reviewer 1 Report

Comments and Suggestions for Authors

High genetic diversity in 3rd generation cephalosporin resistant 2 Escherichia coli in wastewater systems of Schleswig-Holstein

Pathogens is an international, peer-reviewed, open access journal on pathogens and pathogen-host interactions published monthly online by MDPI, with an impact factor of 3.7.

Suggestions

Line 16- ad isolated after 3GCREC.

Line 16 – add the number of 3GCREC that were subjected to whole genome sequencing.

Line 26 – write the conclusion of your study and new findings.

Line 31 - The continuous release of antibiotic-resistant bacteria – add where, in the environment?

Lin 32 – the concept of one health is not just that, please restate the sentence about the connection and how resistance to antimicrobials is included in the concept.

Line 45 - remove the word also or put in the previous sentence that the result was in Germany, the word also was not clear.

Lines 66 and 67 – add why WWTP influent and effluent are increasingly being analyzed for resistance genes and multidrug-resistant bacteria, how these studies can add information in the one health concept.

 Lin 91 – is it possible to add the total amount of sample collected by Buehler 3010 stationary automatic water samplers during the 24 h experiment?

Lines 91 to 96 – is there a reference for the protocol used? Please add.

Lin 96 – please add the macroscopy of the β-glucuronidase-producing colonies in the agar plates.

Line 100 – what is purified?

Line 100 - what is orientational antibiotic susceptibility testing?

Lines 100 to 102 - Why didn't you use cephalosporin discs? explain the reason for choosing the 4 antibiotics and not using cephalosporin discs.

Line 105 – I did not understand how chromID® ESBL agar and the selected disc of antibiotics helped to characterize 3GCREC isolates.

Lines 109 to 111 – is the number of isolates a result? See if this information is a result or methodology.

Line 133 – why did search for point mutations for fluoroquinolones

Line 139/140 - for me you tested only beta-glucuronidase-positive E. coli, not β-glucuronidase positive 3rd generation cephalosporin resistant isolates. Please add information or explain better how you classified as 3rd generation cephalosporin resistant isolates if no disc of cephalosporin was used. Was it by WGS?

Line 149 – maybe the classification of  3GCREC was performed in an previously study? I am a little confused of the 3GCREC classification.

Line 169 – is the percentage correct?

Lines 171 to 173 – remove the sentence.

Line 280 – add some information regarding C. freundii for humans, the importance of the bacteria for humans.

Line 298 – The most common additional phenotypic resistance – in methodology section the characteristic of the 3GCREC must be explained and how the additional and antibiotics were used.

Lin 289 - phenotypically 3GCREC? Again, I did not understand how you can affirm it with the antibiotics disc used. Also, you used chromID® ESBL agar, did you expected to find all isolates to produce a beta lactamase?

Line 290 – did you identified by WGS any of these porins?

Lines 294 to 296 - Is it possible to do some statistical analysis of the relationship between the presence of resistance genes, the phenotypic characteristic, and who has more gene diversity?

Lines 307 and 308 – very important result, that why you need to review the methodology and even the objective of your work, as it appears that you mixed different objectives of a large study. And maybe an analysis of this relationship, see if it is possible. And discuss more if other studies also find this result.

Line 311 and 312 – any reference to reinforce your statement?

Line 326 – was this finding already expected? The diversity?

Discussion section - Your paragraphs in the discussion are very long, try to group similar discussions and have more paragraphs to make reading more fluid and understandable

Line 402 and 403 – were any of the ST identified in your study the first in your country? If so, state that.

After reading your study, I suggest to review the methodology section and to explain why the phenotypic test were performed for the four antibiotics selected, what was the purpose of this analysis. Your work is wonderful and very laborious, so it is worth focusing on what you really want to study, what the real objectives of your study were and what impact they bring and which reinforce the importance of resistance studies with Escherichia coli. Your study provides some important results, focus on that. Be proud of your work, which was not easy and is so important.

Author Response

comment

Answer

Line 16- ad isolated after 3GCREC.

The word „isolated“ was added in the abstract.

Line 16 – add the number of 3GCREC that were subjected to whole genome sequencing.

The number of sequenced isolates was added in the abstract.

Line 26 – write the conclusion of your study and new findings.

A sentence describing the conclusions was added.

Line 31 - The continuous release of antibiotic-resistant bacteria – add where, in the environment?

We have adapted the first sentences of this section in line with the reviewer comments.

Lin 32 – the concept of one health is not just that, please restate the sentence about the connection and how resistance to antimicrobials is included in the concept.

We have adapted the first sentences of this section in line with the reviewer comments.

Line 45 - remove the word also or put in the previous sentence that the result was in Germany, the word also was not clear.

The word “also” was removed.

Lines 66 and 67 – add why WWTP influent and effluent are increasingly being analyzed for resistance genes and multidrug-resistant bacteria, how these studies can add information in the one health concept.

A short explanation was added.

Lin 91 – is it possible to add the total amount of sample collected by Buehler 3010 stationary automatic water samplers during the 24 h experiment?

The total amount and volume of the single samples was added in the text.

Lines 91 to 96 – is there a reference for the protocol used? Please add.

The method was established specifically for this study, so that no direct reference is possible. We had tested the procedure in preliminary experiments in order to obtain single colonies from the different types of water for the production of pure cultures. We have now added a reference to the preliminary experiments.

Lin 96 – please add the macroscopy of the β-glucuronidase-producing colonies in the agar plates.

The red growth of β-glucuronidase-producing colonies on the chromid®ESBL agar plates is now mentioned in the text.

Line 100 – what is purified?

The word was changed to „isolated“.

Line 100 - what is orientational antibiotic susceptibility testing?

The word "orientational" meant that not the full spectrum of antibiotics with potential efficacy against E. coli was tested, but only four substances with high phenotypic discrimination between different isolates. The word “orientational” was removed.

Lines 100 to 102 - Why didn't you use cephalosporin discs? explain the reason for choosing the 4 antibiotics and not using cephalosporin discs.

Cephalosporin discs were used in a double disc diffusion assay. This part of the material and methods section was incomplete and the double disc diffusion assay is now included in the text and in the revised Supplementary Table 3.

Line 105 – I did not understand how chromID® ESBL agar and the selected disc of antibiotics helped to characterize 3GCREC isolates.

Cephalosporin discs were used in a double disc diffusion assay. This part of the material and methods section was incomplete and the double disc diffusion assay is now included. Also isolates with a negative double disc diffusion assay result were further analyzed by Vitek® AST Systems, so that a phenotypic 3rd generation cephalosporin resistance could be confirmed for all isolates included in the sequencing. The Vitek analysis is now included in the material and methods section as well.

Lines 109 to 111 – is the number of isolates a result? See if this information is a result or methodology.

The sentence was switched to the results section.

Line 133 – why did search for point mutations for fluoroquinolones

A short explanation was added.

Line 139/140 - for me you tested only beta-glucuronidase-positive E. coli, not β-glucuronidase positive 3rd generation cephalosporin resistant isolates. Please add information or explain better how you classified as 3rd generation cephalosporin resistant isolates if no disc of cephalosporin was used. Was it by WGS?

All isolates were confirmed to be 3GCREC (see comments above).

Line 149 – maybe the classification of  3GCREC was performed in an previously study? I am a little confused of the 3GCREC classification.

All isolates were confirmed to be 3GCREC (see comments above).

Line 169 – is the percentage correct?

The total number of positive isolates was mistakenly written as “8” instead of “89” and was corrected accordingly. The percentage is correct.

Lines 171 to 173 – remove the sentence.

The sentence in lines 271 to 273 was removed, as it was part of the template.

Line 280 – add some information regarding C. freundii for humans, the importance of the bacteria for humans.

In previous studies on hospital wastewater in a nearby region, β-glucuronidase producing C. freundii were identified in a high proportion among the strains. These studies were included in the text because we wanted to point out that this phenomenon did not occur in the wastewater from the wastewater treatment plants in this study. The C. freundii found in this study were identified in non-remarkable proportions and were not included in the further analysis in any way. We have therefore decided to not include a more detailed description of C. freundii to avoid confusion for the reader and hope for the reviewer's understanding.

Line 298 – The most common additional phenotypic resistance – in methodology section the characteristic of the 3GCREC must be explained and how the additional and antibiotics were used.

It is now clarified in the text, that “additional” meant additional resistance next to phenotypic 3rd generation cephalosporin resistance.

Lin 289 - phenotypically 3GCREC? Again, I did not understand how you can affirm it with the antibiotics disc used. Also, you used chromID® ESBL agar, did you expected to find all isolates to produce a beta lactamase?

All isolates were confirmed to be 3GCREC (see comments above).

As the chromID® ESBL agar contains cefpodoxime as a 3rd generation cephalosporin, we did expect to find a β-lactamase gene in all isolates. Single exceptions of bacteria with other resistance mechanisms than β-lactamases were considered as possibilities as well.

Line 290 – did you identified by WGS any of these porins?

Differential expression of porins was not analyzed in this study, as the main focus of this study was on known genetic markers for resistance.

Lines 294 to 296 - Is it possible to do some statistical analysis of the relationship between the presence of resistance genes, the phenotypic characteristic, and who has more gene diversity?

A statistical analysis of the association between different fluoroquinolone resistance determinants and the expression of a resistant phenotype was performed and included in the manuscript. The statistical analysis confirmed a significant association between the presence of multiple (≥3) point mutations and phenotypic fluoroquinolone resistance.

Lines 307 and 308 – very important result, that why you need to review the methodology and even the objective of your work, as it appears that you mixed different objectives of a large study. And maybe an analysis of this relationship, see if it is possible. And discuss more if other studies also find this result.

A statistical analysis of the association between different fluoroquinolone resistance determinants and the expression of a resistant phenotype was performed and included in the manuscript. The statistical analysis confirmed a significant association between the presence of multiple point mutations and phenotypic fluoroquinolone resistance.

In the course of this analysis, the evaluation of fluoroquinolone resistance was thoroughly rewritten. The results of other studies in which first step mutations were detected were included in the manuscript.

Line 311 and 312 – any reference to reinforce your statement?

The results of other studies in which first step mutations were detected were included in the manuscript.

Line 326 – was this finding already expected? The diversity?

An explanation was shortly added in the discussion part.

Discussion section - Your paragraphs in the discussion are very long, try to group similar discussions and have more paragraphs to make reading more fluid and understandable

Long paragraphs were splitted to make reading more fluid.

Line 402 and 403 – were any of the ST identified in your study the first in your country? If so, state that.

ST15488 could be newly identified in this study and therefore was the first documented finding of this ST. However, we cannot say with certainty that any of the other STs were identified for the first time in Germany in our study. The new identification of ST15488 is now included in the manuscript.

Reviewer 2 Report

Comments and Suggestions for Authors

Dear Authors and Editors,

The article "High genetic diversity in 3rd generation cephalosporin resistant Escherichia coli in wastewater systems of Schleswig-Holstein" addresses the significant One Health concern of multidrug-resistant bacteria transmission, focusing on third-generation cephalosporin-resistant Escherichia coli (3GCREC) through wastewater pathways originating from municipal wastewater treatment plants (WWTPs). The study utilizes whole genome sequencing and multi-locus sequence typing (MLST) to characterize 172 isolates from four German WWTPs. The findings underscore a notable genetic diversity of 3GCREC, highlighting variations in resistance genes, plasmids, and fluoroquinolone resistance-associated point mutations. This emphasizes the intricate dissemination pathways and the limited presence of specific clonal lineages across WWTPs. The article is well-organized and written, with only minor comments as follows:

    Line 43: Change 'E. coli' to 'Escherichia coli (E. coli)'.

    Line 82: Table 1 is missing in the text.

    Line 142: Change 'C. freundii' to 'Citrobacter freundii (C. freundii)'.

    Lines 271-273: Remove this section as it is part of the template.

    Lines 336, 395: Remove double spaces ('than hospital', '. A high').

Good job, congrats to the Authors

Author Response

comment

Answer

Line 43: Change 'E. coli' to 'Escherichia coli (E. coli)'.

E. coli” was changed to “Escherichia coli” in this sentence.

Line 82: Table 1 is missing in the text.

The table is now included as new Supplementary Table 1.

Line 142: Change 'C. freundii' to 'Citrobacter freundii (C. freundii)'.

C. freundii” was changed to “Citrobacter freundii” in this sentence.

Lines 271-273: Remove this section as it is part of the template.

The section was removed.

Lines 336, 395: Remove double spaces ('than hospital', '. A high').

All double spaces in the text were removed.

Round 2

Reviewer 1 Report

Comments and Suggestions for Authors

Thank you for answering my questions. I suggest that when you get the proofread of the manuscript, you check whether any words were spelled incorrectly, or whether there were typos, as the review time is short, but I congratulate you on the speed and quality of the answers and the research.